# Effect of Oils in Feed on the Production Performance and Egg Quality of Laying Hens

**DOI:** 10.3390/ani11123482

**Published:** 2021-12-07

**Authors:** Zhouyang Gao, Junnan Zhang, Fuwei Li, Jiangxia Zheng, Guiyun Xu

**Affiliations:** 1Key Laboratory of Animal Genetics and Breeding of the Ministry of Agriculture, National Engineering Laboratory for Animal Breeding, Department of Animal Genetics and Breeding, College of Animal Science and Technology, China Agricultural University, Beijing 100193, China; gaozhouyangcau@163.com (Z.G.); cauzhangjn@163.com (J.Z.); 2Poultry Institute, Shandong Academy of Agricultural Sciences, Jinan 250100, China; lifuwei1224@163.com

**Keywords:** laying hens, feeding oils, production performance, nutritional composition

## Abstract

**Simple Summary:**

Eggs are a valuable source of protein and fat in the human diet. Due to continuous improvement in the production performance of laying hens, the requirements regarding the feed energy of laying hens are increasing. Oils, which are the main energy sources in feed, exert a substantial effect on the production performance and egg quality of laying hens. In this review, we discuss the classification of oils commonly used in the diets of laying hens and the process of lipid metabolism in laying hens. We further look at the effects of adding different oils on the nutritional composition of eggs with a focus on the mechanism through which dietary oils affect the production performance and egg quality of laying hens, the effects of adding different types of animal and plant oils to the diets, and the potential effects of different types of oils on the health of laying hens.

**Abstract:**

With the development of a large-scale and intensive production industry, the number of laying hens in China is rapidly increasing. Oils, as an important source of essential fatty acids, can be added to the diet to effectively improve the production performance and absorption of other nutrients. The present review discusses the practical application of different types and qualities of oils in poultry diets and studies the critical effects of these oils on production performance, such as the egg weight, feed intake, feed conversion ratio (FCR), and various egg quality parameters, including the albumen height, Haugh units, yolk color, and saturated/unsaturated fatty acids. This article reviews the effects of different dietary oil sources on the production performance and egg quality of laying hens and their potential functional mechanisms and provides a reference for the selection of different sources of oils to include in the diet with the aim of improving egg production. This review thus provides a reference for the application of oils to the diets of laying hens. Future studies are needed to determine how poultry products can be produced with the appropriate proper oils in the diet and without negative effects on production performance and egg quality.

## 1. Introduction

Oils are the most commonly applied sources of energy in feed diets for laying hens and exert multiple effects, such as improving palatability, feed intake, animal immunity, and reducing morbidity [1,2,3]. Compared with that included in the diet of broilers, the amount of oil added to hen feed is low because laying hens have a unique physiological state and are more prone to lipid metabolism disorders than broilers. Therefore, the appropriate proportion and type of oil addition is particularly important for the production performance, lipid metabolism, and egg quality of laying hens.

Eggs have an important economic value, are an excellent source of animal protein, and have become an important consumer product worldwide due to their low cost [4,5]. With the continuous improvement in people’s living standards in developed and developing countries, interest in the internal nutritional composition of eggs, such as the content and proportion of omega-3 and omega-6 unsaturated fatty acids in the egg yolk, is increasing. Eggs consist of the albumen, yolk, and shell. The albumen is composed of 88% water, 11% protein, 0.2% lipids, and 0.8% minerals, and egg yolks comprise 48% water, 17.5% protein, 32.5% lipids, and 2% minerals [6]. The lipids in the egg yolk mainly derive from the oils in the feed, and thus oils play a crucial role in the production performance and egg quality of laying hens. Due to continuous improvements in livestock and poultry production performance, the important role of dietary oils in livestock and poultry feed has attracted increasing attention from breeders and producers. Hence, methods to improve egg quality and enhance egg production performance to meet the diverse needs of consumers have become one of the focuses of the poultry industry.

Currently, the main oils used in the diets of laying hens are vegetable oils and animal oils. The vegetable oils commonly used in hen feeds are rich in a variety of unsaturated fatty acids (UFAs), which are important for the growth and development of laying hens, particularly linolenic acid and linoleic acid, which are essential fatty acids (EFAs) for birds. Linolenic acid deficiency hinders the development of hens and reduces production performance, whereas linoleic acid is deposited directly into the egg yolk and thereby increases the weight of the egg by increasing the weight of the yolk [7]. Animal oils (mainly triglycerides (TGs)) are extracted from animal fat tissue and, when added to the diet, these fats increase the UFA contents of eggs [8,9]. The addition of oils to the diets of laying hens has become an effective method to promote animal growth, increase the FCR, and improve poultry health. The fatty acid composition and physicochemical properties of different types of oils and their physiological effects on laying hens vary substantially [10,11]. As an example, the addition of 6% canola oil will decrease egg production and significantly increase thiobarbituric acid reactive substance (TBARS) levels in eggs on the 21st day of the addition trial (corn–soybean meal compared with 6% rapeseed oil) [12]. Some studies have shown that the addition of 3% and 5% rapeseed oil to the basal diet of 24-week-old Hyland brown chickens has no significant effect on the egg weight [13], but the addition of 2%, 4%, and 6% rapeseed oil to the diet of 40-week-old brown chickens reduces feed intake, egg production, and egg weight [12]. Compared with the results obtained with the addition of 3% fish oil, olive oil, grape seed oil, or soybean oil, the addition of equal amounts of canola oil to the diets of laying hens does not significantly change the egg production, egg weight, feed intake, or FCR but increases the linolenic acid content in the egg yolk [14]. In addition, oils have adhesion properties that reduce dust in hen feed production, promote better particle aggregation, reduce wear on machinery, and result in both low waste and feed savings. The unique aroma of oils may improve the palatability of feeds and enhance their flavor, which leads to increased animal intake. The addition of oils with low heat gain to feed serves to effectively reduce heat stress, which improves feed utilization and reduces mortality [3]. As a result of increasing research on animal nutrition and increasing requirements for animal welfare, environmental protection, and productivity, various properties of dietary oils have been discovered and proven. Therefore, methods to effectively use different types of oils at different concentration ratios according to the nutritional characteristics and production conditions of laying hens are very important in the egg farming industry. This study focuses on the effects of the type, quality, and proportion of oils in the diet on the production performance and egg quality of laying hens and provides a reference for the selection of different types of oils in the egg production process with the aim of improving egg quality and economic efficiency.

## 2. An Overview of Oils

As already mentioned, feed oils are divided into vegetable oils and animal oils. The commonly used vegetable oils in feeds mainly include soybean oil, rapeseed oil, palm oil (PO), and linseed oil, and the animal oils commonly used in feeds mainly include lard, poultry fat, tallow, and fish oil [15]. Different oils have different ratios of saturated fatty acids (SFAs) to UFAs. In general, vegetable oils contain more UFAs than animal oils, which contain more SFAs [16]. Therefore, vegetable oils are currently used at higher levels than animal oils in egg production, but no major difference in energy has been observed. The energy and nutrients that animals obtain from feed depend on the species and age of the animal and vary depending on the quality and chemical composition of the fat [17].

The quality, fatty acid composition, total energy, and price of various oils differ [18]. Various countries and organizations worldwide have established corresponding standards for quality measurements. The main indicators are divided into sensory, technical, and microbial characteristics [19]. The microbial indicators are the total number of colonies and the number of colonies of coliforms, *Salmonella*, *Shigella*, *Staphylococcus aureus*, *Streptococcus hemolyticus,* and other common pathogenic strains [20].

## 3. Mechanism Underlying the Effects of Dietary Oils on the Production Performance and Egg Quality of Laying Hens

### 3.1. Mechanism Underlying the Effect of Oils on Lipid Metabolism in Laying Hens

During the egg-laying period, the amount of lipid mobilization and synthesis in laying hens is in dynamic equilibrium, and the amount of fatty feed added to the diet of laying hens is generally less than 10%. The amount of lipids that laying hens obtain from feed is approximately 3 g per day, and 5–6 g of lipids is needed for the formation of each egg. Therefore, some lipids are synthesized endogenously by laying hens and then deposited in the ovaries through transport via the blood and other circulatory systems to finally form egg yolk material [21]. The fatty acid composition of the diet directly affects the composition of the egg yolk. This observation also serves as the theoretical basis for the inclusion of lipid additives to regulate the egg production performance and egg quality of laying hens. The main organ in which poultry synthesize lipids is the liver, and the fat needed to maintain egg production performance is also mainly synthesized by the liver [22]. The raw materials for synthesizing lipids are fatty acids, glycerol, and cholesterol. These raw materials can be directly supplied by exogenous feed or converted from protein and glucose in the feed. EFAs cannot be synthesized in the body of poultry and can only be obtained from feed [23]. The liver assembles synthetic TGs, cholesterol, and apolipoproteins to form lipoproteins with different diameters and densities, which are responsible for the targeted transport of lipids. Lipoproteins are generally divided into five types according to their density: chylomicrons (CMs), very low-density lipoprotein (VLDL), intermediate-density lipoprotein (IDL), low-density lipoprotein (LDL), and high-density lipoprotein (HDL) [24]. Laying hens have a very strong lipid metabolism in their bodies, particularly during the peak period of egg production. As one of the essential nutrients for animal growth and one of the main components of egg yolk, oils play an important role in regulating the production performance of laying hens and their egg quality. Therefore, the production performance and egg quality of laying hens may be feasibly improved in theory and practice by supplementing the diet with different oils. The development and production of high-quality eggs through dietary oil nutrition regulation technology provides people with foods with balanced fatty acid nutrition, which is not only beneficial to health but also provides a new approach to the production of high value-added poultry products.

### 3.2. Providing Essential Fatty Acids and Affecting Lipid Metabolism

During the peak period of laying hens, the intensity of lipid metabolism in the body increases, and the oils in feed further enter the eggs through the ovaries in the body. The commonly used oils in feed, particularly vegetable oils, are rich in PUFAs (polyunsaturated fatty acids). Most of these fatty acids exhibit an important relationship with the growth and production performance of laying hens. Therefore, fatty acids have become highly valued raw materials in compound feed for laying hens. As hens have physiological requirements for laying eggs, poultry, particularly laying hens, are significantly more dependent on EFA than pigs and ruminants. In particular, linoleic acid and linolenic acid are EFAs that play a decisive role in the growth and reproduction of laying hens [25]. Insufficient dietary EFAs will significantly affect growth and development, affect ovarian development, and reduce the body weight, egg weight, and fertilized egg hatchability [26,27]. Dietary supplementation with linoleic acid increases the egg weight [28,29]. In addition, when the diet of laying hens meets the demand for linoleic acid, the addition of exogenous oil significantly increases the egg weight independently without affecting the dietary metabolizable energy and the linoleic acid content [30]. Vegetable oils rich in PUFAs reduce the liver fat content or alleviate fatty liver syndrome in laying hens, which is conducive to lipid metabolism in laying hens, whereas excessive consumption of animal oils or starch aggravates fatty liver syndrome in laying hens or significantly increases hepatic lipid deposition in laying hens [31,32].

### 3.3. Effect of Oils on the Nutritional Composition of the Yolk

Due to improvements in people’s living standards, people are currently paying increasing attention to the nutritional content of eggs. The UFA content is an important basis for determining the nutritional value of poultry eggs. A certain amount of UFAs in the diet exerts a positive effect on animal health [33]. The lipids of the egg contain SFAs (30–35%) and UFAs (30–33%) and, among these, 0–45% and 20–25% of UFAs are MUFAs and PUFAs, respectively [34]. In addition, the fatty acid composition of the egg yolk substantially affects the flavor of the egg [35]. The use of feed rich in n − 3 fatty acids increases the content of n − 3 fatty acids in eggs. Therefore, the fatty acid composition and content of egg yolk can be adjusted through the feed formula to produce eggs rich in PUFAs [36]. Eicosapentaenoic acid (EPA, C20:5n − 3) and docosahexaenoic acid (DHA, C22:6n − 3) are both n − 3 PUFAs, and supplementation with EPA and DHA significantly improves brain and cardiovascular function [37,38]. Eggs rich in n − 3 PUFAs have been sold in the United States, Canada, Australia, and other countries and have shown excellent nutritional benefits [39]. Published studies have confirmed the effects of oils on the fatty acid composition of egg yolks and their promotion of human health [40,41,42,43].

After the addition of suitable oils, the composition and proportion of FAs in the yolk will significantly change. FAs mainly contain SFAs, MUFAs, and PUFAs. Egg yolks are rich in PUFAs, which are divided into ω − 3 and ω − 6 PUFAs. Among ω − 3 PUFAs, those that exert the most significant effects on the human body are DHA and EPA [44]. Studies have shown that both ω − 3 and ω − 6 fatty acids exert anticancer effects, and the anticancer effect of ω − 3 polyunsaturated fatty acids is significantly better than that of ω − 6 polyunsaturated fatty acids [45]. In addition, the ideal intake ratio of ω − 6 PUFAs and ω − 3 PUFAs would be between 4:1 and 10:1 [46]. The method for increasing the content of specific unsaturated fatty acids in eggs by adding unsaturated fatty acids to the diet of laying hens has become increasingly mature. People use laying hens as a fatty acid converter to achieve nutritional optimization. For example, the addition of 5% rapeseed oil to the diet of laying hens significantly increases the weight of the egg yolk and the contents of DHA and n − 3 FAs in the egg yolk [13]. Han et al. [47] showed that the addition of 1.5 to 4.5% sesame oil to the diet increases the UFA content in egg yolks and reduces the blood lipid levels of laying hens.

### 3.4. Immunomodulatory Effect of Oils

Fatty acids in oils play an important role in maintaining poultry health. The classic indices include ∑SFAs, ∑MUFAs, ∑PUFAs, ∑n − 6 PUFAs, ∑n − 3 PUFAs, and n − 6 PUFAs/n − 3 PUFAs [48]. When the body is exposed to external antigens, the secretion of lymphokines and antibodies and the production of new immune cells depend on the participation of fat [49]. PUFAs in the diet are involved in the synthesis of biofilm structures and are precursors for a variety of physiologically active substances, such as eicosanoids and leukotrienes [50]. These substances are particularly important for poultry and may participate directly or indirectly in physiological processes, particularly immune system processes [51]. PUFAs significantly improve the inflammatory response in the body and reduce the serum TNF-α and IL-1β levels [52]. The intake of PUFAs by laying hens significantly increases their serum lysozyme activity and improves immune function [53]. Peroxisome proliferator-activated receptors (PPARs) are nuclear hormone receptors and transcription factors. Long-chain UFAs may mediate lipid metabolism and the immune inflammatory response in laying hens through the PPAR pathway [54]. In addition, the FAs in the egg yolk are mainly synthesized by the liver, which is substantially affected by the fatty acid composition of the diet. The addition of linoleic acid and linolenic acid to the diet of laying hens significantly increases the contents of arachidonic acid and DHA in the egg yolk [55]. Various PUFAs, such as linolenic acid and arachidonic acid, are components of the cell membrane structure and are essential for maintaining the integrity of the cell membrane structure and function. In addition, EFAs are precursors of eicosanoids, such as prostaglandins, prostacyclins, thromboxane, and leukotrienes, and these substances are used for blood coagulation, nerve signal transduction, embryonic development, reproduction, immune response, and bones in poultry. These compounds play an important role in physiological processes such as embryonic development and are also involved in the transport of body fluids, the activation of certain enzymes, and the metabolism of lipids, particularly cholesterol, which are very important for maintaining and improving the production performance and egg quality of laying hens [1,56].

In modern egg production, laying hens are usually under high-intensity production pressure. Heat stress and oxidative stress (OS) are common factors that affect the production performance of laying hens [57,58]. The digestion and absorption efficiency of oil is outstanding, and its heat increment is significantly lower than that of carbohydrates and protein. This property may substantially reduce the heat dissipation burden of the animal’s body in summer, particularly in high-density laying hen production, and thereby alleviate heat stress and improve the FCR [59]. The double bond in oil compounds exerts a protective effect on easily oxidized substances in the body (such as vitamin E), which might improve the body’s antioxidant capacity and relieve OS. Therefore, the benefits of oils for animal health are likely attributed to their effects on the immune and antioxidant systems in the body [60].

### 3.5. Effects of Dietary Oils on the Structure and Function of Intestinal Microbes

The structure and function of the gut microbiota are critical to the health and production performance of laying hens. The acquisition and establishment of the intestinal microbiota throughout the poultry production cycle exerts a substantial effect on the development and physiological regulation of the intestine and maintains intestinal homeostasis (i.e., nutrition, metabolism, immunity, and integrity of the intestinal barrier), which results in the optimization of the host’s energy absorption and use efficiency [61]. The processes of the digestion and absorption of nutrients are closely related to the intestinal microbiota, and the nutrient absorption, FCR, and production performance of the host are affected by the composition and diversity of the microbiota [62,63]. The microbial composition of the poultry intestine is affected by many internal and external factors, such as the host and environment. Feeding and nutrition management are important factors that affect the composition and function of intestinal microbes in poultry. Therefore, the intestinal microbiota can be adjusted by regulating the feed composition [64]. At the phylum level, the gut microbiota of chickens includes hundreds of species, including microbial species dominated by *Firmicutes, Bacteroides*, *Proteobacteria, and Actinomycetes* [65,66]. Oil additives such as α-linolenic acid (ALA), DHA, and glycerin and its derivatives (such as glycerol monocaprylate and glycerol monolaurate) exert a positive effect on the intestinal microbial composition, egg production performance, egg quality (e.g., egg weight and egg yolk FA composition), reproductive performance, and body health of laying hens, which indicates the potential role of dietary oils in improving the microbial community, lipid metabolism, and health of laying hens [67,68,69,70]. For example, studies have found that the addition of linseed oil and algae oil to the diet increases the types of Firmicutes (such as *Faeculus*, *Clostridium,* and *Ruminococcus*) microorganisms, and these microorganisms are closely related to FA metabolism [68,69]. Different types and amounts of lipids affect the health of the host to different extents, which may be due to their varying effects on the composition of the gut microbiota [71,72].

## 4. Relationship between Fatty Acids in Human Diets and Health Indicators

Studies have found that eating the right amount of egg every day can reduce the risk of metabolic syndrome in adults over the age of 40 and has a significant positive effect on blood glucose and triglyceride levels in men. The reason for this may be that fatty acid composition has an immense impact on the dietary factors of fats and oils. The quality of dietary fat mainly depends on the ratio of n − 6 to n − 3 fatty acids [73]. The fat content in the human diet has related effects on common diseases such as the risk of coronary heart disease (CHD) and type 2 diabetes [74]. The thrombogenic index (TI) and atherogenicity index (AI) are the most commonly used indices to assess the composition of FAs. The AI and TI were proposed by Ulbritcht and Southgate in 1991 [48], where the AI represents the sum of SFAs and the relationship with the sum of unsaturated fatty acids (UFAs), which also represents the atherosclerotic potential of saturated fatty acids; the TI describes the thrombotic potential of FAs, predicts the tendency of thrombus formation in blood vessels, and indicates the functions of different FAs. At present, AI indicators have been widely used to evaluate the quality of meat, eggs, dairy products, the nutrient composition of oil, and other livestock products because they are not only crucial for the nutritional value of eggs but they also affect shelf life. In addition, the TI was used due to the correlation between FAs and human health [49,75].

## 5. Effect of Different Oils in Diets on the Production Performance and Egg Quality of Laying Hens

The use of specific dietary oils depends on the price and availability of feed ingredients. As mentioned above, different dietary oils affect the production performance and egg quality of laying hens by regulating the lipid metabolism, immune function, and intestinal microbial composition, and this phenomenon provides a theoretical basis for the oil-mediated regulation of the laying performance and egg quality of laying hens. However, due to differences in the physical and chemical properties, lipid composition, main functional substances, quality, and added amount of different types and sources of oils, their effects on the health, laying performance, and egg quality of laying hens are also different. In general, for a single oil, the digestion and absorption of vegetable oils is significantly better than that of animal oils, but balanced and combined oils are better than single oils. Table 1 lists the effects of animal and vegetable oils on egg quality, production performance, and nutrients.

### 5.1. Vegetable Oils

#### 5.1.1. Soybean Oil

Soybean oil has a low production cost and easy quality control and occupies an important position in global vegetable oil production and consumption. In this context, soybean oil has become a preferred fat additive for laying hens. Soybean oil is a high-quality source of fat in poultry farming, particularly for the growth and development of chicks [135]. Among soybean oil FAs, linolenic acid accounts for 50% to 55% of the total fat content, and oleic acid accounts for 20%; in addition, soybean oil contains a large amount of UFAs. Soybean oil has a higher EFA content than other vegetable oils and, thus, the use of soybean oil in the diet of laying hens exerts a positive effect on production [136,137]. Due to differences in soybean raw materials and processing, the energy and nutritional value of different soybean oils varies substantially.

Numerous studies have shown that the addition of soybean oil to the diets of laying hens has no adverse effects on the egg production rate and feed-to-egg ratio; this oil also effectively relieves heat stress in laying hens under high-temperature conditions and increases the egg production rate and FCR [76,77,78,79,138]. The addition of 1% soybean oil to the diet may increase the serum alkaline phosphatase (ALP) content, thereby promoting calcium and phosphorus metabolism in the body, increasing the amount of calcium deposition in the eggshell, and significantly reducing the rate of soft-shelled eggs [76]. Soybean oil has no significant effect on the serum lipid content of laying hens, but this oil tends to reduce the serum TG, total cholesterol (T-CH), and LDL cholesterol (LDL-CH) contents, increase the concentration of HDL cholesterol (HDL-CH), and improve glucose and lipid metabolism in laying hens [76]. Moreover, the addition of soybean oil to the diet under high-temperature conditions increases the total antioxidant capacity (T-AOC) in serum and reduces the content of malondialdehyde (MDA), which improves the antioxidant capacity of laying hens [77,79].

The addition of soybean oil to feed improves the nutritional value of the eggs without negatively affecting other aspects of egg quality. In general, the addition of soybean oil to the diet of laying hens increases the levels of linoleic acid and n − 3 and n − 6 PUFAs in the eggs, enriches the types of n − 3 fatty acids in the yolk, and improves the color of the yolk but has no negative effects on the albumen weight, albumen height, or HU. Moreover, the dietary addition of soybean oil has no negative effects on the eggshell thickness, eggshell strength, or other eggshell quality indicators, but lightens the eggshell color [10,11,13,40,76,77,79,80,81,139,140,141]. The egg weight may mainly depend on the type of oil fed to laying hens; however, the results of different studies on the effect of soybean oil on egg weight traits are not consistent [10,139,142]. The content of n − 3 PUFAs largely determines the dietary value of eggs [143]. In addition, the function of PUFAs depends on the ratio of n − 3 to n − 6 PUFAs, and this ratio has important nutritional value [144]. Dietary supplementation with soybean oil increases the n − 3/n − 6 PUFA ratio, and this increase is mainly due to an increase in the n − 3 PUFA content [139,145]. In addition, the use of soybean oil reduces the cholesterol content in eggs and reduces the TI and AI, which confirms the nutritional and health value of soybean oil in the diet of laying hens [81,139].

#### 5.1.2. Rapeseed Oil

Rapeseed is the second largest oil crop worldwide. It is planted in broad areas in China, Canada, Europe, India, and Australia, and the planted area and total output of rapeseed in China account for approximately 20% of the global rapeseed planting area [146]. Rapeseed oil, an edible vegetable oil rich in UFAs, plays an important role in daily life [147,148]. In addition to containing a certain amount of oleic acid, linoleic acid, and linolenic acid, rapeseed oil also contains erucic acid. In rapeseed oil and other cruciferous plant seed oils, erucic acid accounts for approximately 20–55% of the fatty acid content, which is an important factor affecting the feeding value and nutritional value of rapeseed oil [149].

Research on the effect of rapeseed oil on the performance of laying hens has mainly focused on different levels of oil addition, but very few reports on the effect of rapeseed oil with different erucic acid contents on laying hens have been published. As rapeseed oil is rich in erucic acid and contains a large amount of UFAs, the addition of very high levels of rapeseed oil to the diet will reduce the performance of laying hens [82,150,151]. Therefore, the results on the effect of rapeseed oil on the performance of laying hens are not consistent and further study is needed.

The addition of 1–5% rapeseed oil to the diets of laying hens has no significant effect on the egg production rate, total egg production, egg weight, feed intake, body weight gain, or FCR [11,13,83,152,153]. However, studies have shown that the addition of 2–6% rapeseed oil to the diet of laying hens reduces egg production, egg weight, feed intake, and the feed-to-egg ratio [12,82,150]. In general, the addition of 2–6% rapeseed oil to the diet of laying hens exerts a limited effect on conventional parameters of egg quality, such as albumen weight, egg yolk weight, albumen height and, therefore, HU [12,13,83,84,150,153]. The addition of rapeseed oil might increase the lipid content and composition of the egg yolk and increase the contents of oleic acid, linolenic acid, DHA, and total n − 3 PUFAs in eggs, but it does not significantly change the cholesterol content [11,13,150].

Adding rapeseed oil to the diet significantly reduces the serum TG content and increases the HDL-C content in laying hens. A diet including rapeseed oil with a low amount of erucic acid reduces the serum lipid content and liver fat deposition in laying hens, and these effects may be related to the increases in the expression of the apolipoprotein synthesis genes apoB-100 and apoBVLDL-II observed with this diet, which lead to increased lipid transport. In addition, the levels of PUFAs and MDA in the eggs of hens fed diets containing rapeseed oil with low levels of erucic acid are higher and lower, respectively, than those of hens fed diets with high levels of erucic acid. Moreover, the deposition of erucic acid in egg yolk is strongly correlated with the content of erucic acid in rapeseed oil, which indicates that erucic acid increases the level of lipid peroxidation in eggs [150].

#### 5.1.3. Linseed Oil

Linseed is an annual herbaceous plant with seeds that produce a high oil content. It is an edible oil obtained from flaxseeds through low-temperature cold pressing to acquire crude oil that is then refined [154]. Linseed is one of the top ten oil crops worldwide, and flaxseed oil contains a variety of UFAs, such as ALA, linoleic acid, and oleic acid. The ALA content can be as high as 53%, which results in a high nutritional value in promoting cholesterol metabolism. In recent years, the addition of flaxseed oil to animal feed has been shown to improve production performance, reduce the feed-to-egg ratio, and significantly increase the nutritional value of feed [155]. However, the addition of a reasonable amount of linseed oil to the diet of laying hens of different breeds under breeding conditions used in actual production and application still requires further study. Research on its potential physiological functions, such as promoting an increase in enzyme activity and the proliferation of intestinal probiotics, remains in the exploratory stage.

The addition of 2 to 3% linseed oil to the diet of laying hens induces no significant changes in production performance, such as egg production, egg weight, feed intake, FCR and live weight, egg quality, or eggshell quality [11,85,86,105,156,157]. However, due to the presence of antinutritional factors in flaxseed, laying hens fed a diet containing 5% or higher concentrations of flaxseed oil will exhibit a significantly reduced body weight and significantly decreased egg production [87,106,151]. Therefore, relatively low concentrations of linseed oil should be explored and fed to laying hens to determine their effect on laying hen performance and egg quality.

Linseed oil is one of the richest sources of ALA and has a high n − 3 PUFA deposition efficiency. It has been widely used in the diets of laying hens to produce n − 3 PUFA-enriched eggs [88,106,156,158]. The beneficial effects of n − 3 FAs on health are mainly attributed to EPA and DHA. As linseed oil contains a high ALA content, EPA and DHA can be synthesized from ALA and deposited in the egg during metabolism [89,90]. The addition of linseed oil to the diet of laying hens increases the content of n − 3 PUFAs, such as oleic acid, linolenic acid, EPA, and DHA, and the deposition of MUFAs decreases as the level of linseed oil increases; however, the change in the cholesterol content observed in different studies is inconsistent [11,13,36,91]. The enrichment of n − 3 PUFAs in egg yolks leads to a deterioration of the organoleptic properties of eggs, which affects the acceptance and preference of consumers [159] and may be one of the main factors limiting the production of n − 3 PUFA-enriched eggs. According to reports, the addition of 5% flaxseed to the diet of laying hens has little effect on the organoleptic quality of eggs. However, as the level of supplemented flaxseed oil increases, the abnormal flavor produced by n − 3 PUFA enrichment increases, resulting in a decrease in egg acceptance [86,92]. The eggs obtained from laying hens fed more than 5% linseed oil have characteristic flavors that have been described as fishy, tuna, and ocean-like [93,160]. In summary, linseed oil is an ideal source of ALA, which can be used for the production of n − 3 PUFA-enriched eggs, but the effect of long-term feeding with linseed oil on laying hens requires further research. Moreover, changes in the sensory properties and consumer acceptance should be considered when producing n − 3 PUFA-enriched eggs.

#### 5.1.4. Palm Oil

PO (Palm oil), which is one of the main edible oils worldwide, is obtained by pressing mature PO pulp. The global annual output of PO exceeds 50 million tons, which accounts for more than 30% of the total global oil production. The total output and consumption of PO exceeds that of soybean oil and is ranked first, and approximately 85% of PO is used in food applications. PO contains balanced saturated and UFA esters and is composed of 50% SFAs (mainly palmitic acid), 40% MUFAs (mainly oleic acid), and 10% PUFAs (mainly linoleic acid). In addition, PO is rich in a variety of healthy phytonutrients, such as vitamins, tocotrienols, carotenoids, phytosterols, phospholipids, and polyphenols. Although the contents of these trace components are less than 1%, they play an important role in the stability and quality of PO, which makes PO a balanced and nutrient-rich source of fat [94,161,162,163].

Palm trees are mainly planted in West Africa and Southeast Asia. The bright red oil extracted from palm fruit is called crude palm oil (CPO) [164]. Studies have also shown that CPO is particularly rich in natural vitamin E and may confer specific health benefits, which may improve the palatability of hen diets [165]. CPO is subjected to refinement treatments such as free fatty acid removal, decolorization, and deodorization to obtain salad-grade refined PO (refined, bleached, and deodorized PO and red PO (RPO)) and meet market demand. While PO does not contain the impurities and odor compounds found in CPO, it also no longer contains carotenoids and vitamin E [161]. The improved refinement process removes free fatty acids and odor compounds while retaining the greatest possible amounts of nutrients, such as carotenoids, which results in the production of RPO [162,163].

In general, as the amount of added RPO (1–3%) increases, the production performance of laying hens increasingly improves. Importantly, the color of the egg yolk, in addition to its nutritional value, is significantly improved by the addition of RPO to the diet. This oil also improves the flavor of the egg.

The addition of PO to the diet does not negatively affect the production traits of laying hens (e.g., feed intake, average daily gain, FCR, and egg production rate) or conventional egg quality indicators (HU, albumen weight, egg yolk index, egg shape index, eggshell quality, and eggshell color) [95,96,166]. In particular, studies have noted that increases in the CPO content in the laying hen diet are associated with improvements in the production performance of the laying hens (feed intake, FCR, and egg laying rate) and egg quality (egg weight, egg yolk weight, HU, egg yolk index, and eggshell quality) [97,98,166]. The addition of RPO has no effect on the feed intake and feed-to-egg ratio of laying hens, and the laying rate obtained with 3% RPO group is 4.77% lower than that obtained with 1% RPO [167].

RPO reduces the TG levels in the egg yolk and significantly increases the MUFA contents in the egg yolk; the PUFA content decreases with the increase in the RPO content in feed, whereas the SFA content remains unchanged [96,167]. As PO is rich in carotenoids and lutein, feeding PO to laying hens can significantly deepen the egg yolk color and reduce the degree of lipid peroxidation in the yolk. The color grade of egg yolks and the contents of various carotenes vary depending on the RPO content in the feed, and an increase in color grade is observed with increases in the RPO content [96,97,99,100,166,167]. CPO is a rich source of tocotrienols, which inhibit the biosynthesis of cholesterol. According to previous studies, when laying hens are fed a 3% CPO diet, the yolk cholesterol concentration is reduced by 36% (18.6~11.9 mg/g) and the yolk content is also increased. The depositions of vitamin E, α-tocotrienol, and tocotrienol in the egg yolk are potential sources of antioxidants and cholesterol-lowering agents in the human diet [99,166].

#### 5.1.5. Cottonseed Oil

Cottonseed oil (CSO) is mainly composed of linoleic acid (52%), palmitic acid (24%), and oleic acid (22%) [101] and is also rich in fat-soluble vitamins; CSO is often used as an energy source in feed. In general, CSO is a high-quality edible vegetable oil with a high nutritional value and low cost. This oil meets the national edible oil standard after refinement and is also widely used in the poultry industry. However, cottonseed contains free gossypol (FG), cyclopropylene fatty acid (CPFA), and other toxic and harmful substances [168,169] that inevitably remain in cottonseed cake, meal, and CSO after processing, which limits the use of this oil in livestock and poultry feed. In particular, due to the addition of excess CSO to the diets of laying hens and poultry, CSO can significantly affect the liver metabolism of laying hens, leading to changes in the hepatic expression of genes in important pathways such as metabolism, immunity, and signal transduction. CSO may also be enriched in the egg yolk through blood transport, leading to an increase in the HDL/LDL ratio in the egg yolk and a change in the ratio of fatty acids in the egg yolk, which results in a hardening of the egg yolk and the formation of a “rubber egg”. Mu et al. [101] showed that the addition of 2% CSO to the diet of laying hens could harden the yolk and produce “rubber eggs”, thereby affecting the taste of the eggs; the addition of CSO to the diet (58 mg/kg ≤ CPFA ≤ 116.0 mg/kg) slightly increases the yolk color and significantly reduces the laying rate, egg weight, and FCR of laying hens. Traditional egg quality indicators, such as the egg yolk index, HU, and albumen height, do not effectively represent the egg quality of rubber eggs, which illustrates the limitations of traditional egg quality indicators. Laying hens are particularly sensitive to gossypol, and as this accumulates in the body, the tissues are damaged to varying degrees. This compound also damages the reproductive system and liver of hens and may also alter the levels of serum biochemical indicators [102,168,170]. Chelating compounds formed by the reaction of gossypol with protein in the diet may reduce the digestibility of protein and iron and exert an adverse effect on the growth performance of laying birds. In addition, the toxic effects of cotton phenol are growth inhibition, lameness, reduced egg size, and hatchability. CPFA also causes pink discoloration of yolk spots and albumen, thus deteriorating the internal quality of the eggs [168]. The FG content is related to adverse reactions such as decreased production performance, a decreased FCR, discoloration of the egg yolk, and decreased hatchability [88,89,90,91,92,93,94,95,96,97,98,99,100,101,102,103,159,160,161,162,163,164,166,167,168,171]. Therefore, the EU and other countries have issued standard limits for the FG content in various feed ingredients and different animal feed products, and the standard stipulates that the amount of FG added to laying hen diets should not exceed 20 mg/kg. In poultry production, particularly laying hen production, the amount of CSO added to the diet should be strictly controlled, and the content of FG in the diet should be controlled to avoid unnecessary losses. Decreases in egg production performance and egg quality caused by feeding cottonseed meal to hens may not be caused by FG but by the high content of arginine in the cottonseed meal. In addition, as a limiting amino acid, a low lysine content is one of the factors that affect the use of cottonseed meal, and this effect might be mitigated by adding lysine to the diet [104].

#### 5.1.6. Microalgae Oil

As a producer of n − 3 PUFAs and an excellent source of animal feed, microalgae have received extensive attention in recent years [172,173]. DHA resources have attracted increasing attention; DHA in fish oil is not primarily synthesized within fish but is mainly produced by marine microalgae. Therefore, microalgae as a source of DHA have become a prominent research topic. *Schizochytrium* sp., *Ulkenia amoeboida,* and *Crypthecodinium cohnii* were approved for the production of DHA algae oil, and their addition to foods is allowed. N − 3 PUFAs, particularly DHA, play an important role in lipid metabolism and the synthesis of biologically active molecules. These PUFAs ensure the normal physiological functions of cells, promote the development of the retina and brain, delay the aging of the brain, prevent and treat cardiovascular diseases, and exert antitumor and anticancer effects. N − 3 PUFAs also regulate the immune function of the body and inhibit inflammation, which exerts positive effects on the growth and production performance of humans and animals [174].

Microalgae are rich in UFAs, vitamins, carotenoids, and minerals. In microalgae, oil accounts for more than 70% of the dry cell weight, and the total fat contains 35–40% DHA. More than 90% of fatty acids are present in the form of TGs, which are easy to separate and purify and do not contain the fishy smell of fish oil [175,176]. The carotene in microalgae is not only a natural antioxidant but also deposited in the egg yolk to deepen the yolk color [177]. Microalgae have many advantages. Bioreactors using light microalgae have a certain resistance to marine pollutants, reducing the deposition of marine heavy metal pollution in the body. They also artificially optimize the growth environment, and the EPA content in microalgae fat particles is lower than that in fish oil, which reduces the competitive inhibition of amino acids [178].

The addition of microalgae/microalgae oil to the diet does not negatively affect the production performance (e.g., egg production, egg weight, feed intake, and FCR) and egg quality (external quality and eggshell quality) of laying hens [105,107,108,109,110,111,112,113,114,157,179,180,181,182,183]. The addition of microalgae or algae oil to the diet of laying hens substantially increases the deposition of DHA, EPA, and n − 3 PUFAs in the egg yolk and decreases the deposition of total n − 6 PUFAs, which increases the ratio of n − 3 to n − 6 PUFAs in the egg yolk. The main type of UFA in microalgae oil is DHA-TG, and its absorption efficiency is approximately 50%; the amount of DHA deposition in microalgae oil increases with increases in the level of microalgae addition. In addition, based on the premise of equal levels of DHA in the diets, the addition of choline increases the DHA concentration in eggs, and microalgae oil affects the total lipid content of the egg yolk and plasma but does not affect the quality or cholesterol content [105,107,111,112,113,114,115,116,117,118,119,120,157,180,181,182,183]. When both microalgae oil and choline are added, the DHA-TG provided by *Schizochytrium* oil is converted into DHA-phospholipids in laying hens through acyl exchange and transesterification. Moreover, choline provides additional substrates for the synthesis of DHA-PC through the phosphatidylcholine pathway and synthesizes more DHA-phospholipids and DHA-lysophosphatidylcholine, which are enriched in the egg yolk, and these effects promote DHA enrichment in the egg [115]. As DHA inhibits the key enzyme (3-hydroxy-3-methylglutaryl-CoA reductase) that synthesizes cholesterol in the liver, it leads to restricted cholesterol synthesis and lower blood cholesterol levels [121]. In addition, dietary microalgae oil supplementation promotes the deposition of β-carotenoids in the yolk and improves the yolk color [109,112,114,184]. However, when algae oil is used as the source of n − 3 PUFAs for obtaining enriched eggs, the resulting changes in the flavor, taste, and texture characteristics of eggs leads to a decrease in egg acceptance. The addition of algae oil (≥2.5%) to the diet results in significant decreases in the flavor and acceptance of eggs [105,159]. Therefore, the effect of changes in the fatty acids in the egg yolk on the flavor of eggs should be considered in the production of n − 3 PUFA-enriched eggs.

Microalgae oil is rich in DHA, and dietary supplementation with microalgae oil can reduce the serum TC, TG, and cholesterol levels in laying hens, increase the VLDL and HDL levels, and increase the transport of DHA to oocytes. In addition, microalgae oil increases the levels of H9 and NDV antibodies after immunization, significantly increases the relative expression of IL-2, IL-10, and IFN-γ genes in the spleen, reduces the degree of inflammation, promotes the recovery of cellular immune function, and improves the lipid metabolism and immunity of laying hens [115,180,185].

### 5.2. Animal Oils

#### 5.2.1. Fish Oil

Fish oil is a traditional source of n − 3 PUFAs, containing more than 30% n − 3 PUFAs [186]. It contains relatively few SFAs, and the ratio of n − 6 to n − 3 PUFAs is approximately one. Studies have shown that feeding hens a diet containing 3% fish oil significantly increases egg production and egg weight [187]. However, the addition of fish oil to the diet may affect the performance of laying hens, and the most common change is a decrease in the egg yolk weight. Lawlor et al. [188] reported that the addition of 2% microencapsulated fish oil to the diet leads to a significant reduction in the egg yolk weight. However, other studies found that feeding fish oil to laying hens did not significantly change the egg weight [107,122]. Excess fish oil exerts potentially harmful effects on animals by increasing the sensitivity of the heart and liver mitochondria to lipid peroxidation [123]. The production performance and egg quality of laying hens are reportedly significantly reduced by feeding excess fish oil to hens. The addition of 8% fish oil to the diet significantly reduces the feed intake, egg production, FCR, egg weight, egg yolk color, egg yolk weight, and eggshell color without affecting the eggshell thickness. Relatively high values for the albumen height and HU might be associated with a smaller yolk, and significant increases in the blood MDA content, aspartate transaminase (AST) activity, and uric acid (UA) content indicate that excessive dietary fish oil induces lipid peroxidation and impairs liver function, whereas a decrease in shell color is associated with a reduced synthesis of protoporphyrin IX [79].

Fish oils are rich in EPA and DHA. Feeding fish oil to laying hens can increase the deposition of EPA and DHA in the egg yolk. The addition of different fish oils to the diets of laying hens significantly increases the n − 3 PUFA contents in the egg yolk, reduces the total n − 6 PUFA content in the egg yolk, and increases the n − 3/n − 6 PUFA ratio [119,124,140,187,189]. Laying hens convert EPA into DHA and deposit the latter in the egg yolk. The main n − 3 PUFAs deposited in the egg yolk are DHA and a small amount of EPA. The EPA conversion efficiency is slightly lower than that in the diet, and DHA is directly deposited in the egg yolk [140]. The amounts of EPA and DHA deposited in the egg yolk increase as the amount of added fish oil increases, but the deposition efficiency continues to decrease. However, when 1.5% fish oil is added to the diet, the eggs have a fishy smell that is not easily accepted by consumers. Eventually, fish oil will affect the egg quality of n − 3 PUFA-enriched eggs [188]. EPA and DHA are preferentially deposited in phospholipids, which are structural lipids, in the egg yolk and may be one of the reasons for the observed decrease in the deposition efficiency [140]. Although an increase in the level of fish oil in the diet significantly increases the n − 3 PUFA content in eggs, fish oil will be subjected to many restrictions if used as the raw material for n − 3 PUFA production. For example, and most importantly, fish oil resources are limited. Second, fish oil is easily oxidized. If used improperly, the fishy smell of eggs will be aggravated. Therefore, poultry farmers should pay attention to the amount of fish oil added to feed. Although the treatment of fish oil by microencapsulation technology better eliminates the resulting fishy taste in the feed, it has little effect on improving the fishy smell of the resulting eggs [125]. In addition to the fishy smell of eggs, fish oil may also contain persistent organic pollutants (POPs). Currently, the number of fish in the ocean is only 10% of the population that existed during the preindustrial revolution, and this number is thus no longer sufficient to provide a source of DHA supplements. Therefore, DHA sources other than fish oil are now being gradually developed.

#### 5.2.2. Lard

Lard, which is derived from pigs, is obtained from the rendering of the adipose tissue of pigs and is often used together with other vegetable oils to produce margarine and other specialty food oils [190,191]. Lard is mainly composed of saturated fatty acids (40–45%). For example, palmitic acid and stearic acid are present at high levels (25% and 15%, respectively). Lard also contains UFAs: the highest content of MUFAs is approximately 40–45% and PUFAs, which mainly include linoleic acid, constitute 10–15%.

As an important fat added to feed, lard contains a variety of fatty acids and UFAs and can provide higher calories; therefore, it is also widely used in layer feed. However, studies examining the effects of lard on hen performance and egg quality have also reported different results. At present, some studies have found that adding lard to feed improves the egg quality to a certain extent. For example, adding lard increases the level of SFAs in the egg yolk and significantly improves the yellow color of the egg [126,127]. Areerob et al. [166] found that the addition of 2% lard to 49-week Hessex layers had no significant effect on the hens’ egg production rate, feed intake, or FCR compared with the results found for the control group. However, many studies have shown that adding vegetable oils can reduce the liver fat content or relieve fatty liver syndrome in laying hens, whereas animal oils can aggravate fatty liver syndrome in laying hens or may have a negative impact on the lipid metabolism of laying hens [32]. The addition of 3% lard to the diet will significantly decrease the egg production rate, egg weight, and egg hatchability and fertility of laying hens compared with the results found with the control diet (corn–soybean meal) [128].

#### 5.2.3. Cod Liver Oil

Cod liver oil is one of the oldest nutritional supplements in the world and a type of marine oil that has been used since 1789. From a nutritional point of view, this oil contains high amounts of EPA and DHA from omega-3 unsaturated fatty acids as well as high amounts of vitamins A and D [192]. On a medical level, supplementation with high doses of cod liver oil can be used to treat individuals suffering from vitamin D deficiency and is also commonly used in the treatment of rickets. Thus, cod liver oil plays a role in egg production by supplementing hens that have vitamin D3 deficiency and promoting the absorption of calcium in their bodies [193].

Many studies have shown that cod liver oil can have an impact on the nutritional composition of eggs and the antioxidant index of hens. The addition of cod liver oil to the diet of laying hens can significantly reduce the plasma cholesterol levels [129]. Tu et al. [194] found that supplementation with 3% cod liver oil reduces the serum levels of triglycerides, cholesterol, and LDL-C+ VLDL-C and thereby reduces the cholesterol level in the egg yolk. Hargis et al. [130] also noted that supplementation with cod liver oil changes the ratio of unsaturated fatty acids in the egg yolk, e.g., supplementation with 3% marine oil increases the C20:5 and C22:6 contents of egg yolk lipids. However, because cod liver oil is more expensive than other fats and oils, the quality of cod liver oil on the market varies, and the use of poor-quality cod liver oil can deteriorate the quality of eggs; moreover, the addition of a too-high amount of cod liver oil can cause eggs to have a distinct, fishy smell [131]. Therefore, attention should be paid to the level of supplementation and the quality of the added oil during the addition process.

### 5.3. Balanced Oil

As mentioned above, different types of oils contain different main fatty acids; for example, PO is rich in palmitic acid and a variety of healthy phytonutrients, and soybean oil has the highest content of linolenic acid. Thus, a variety of oils can be blended to compensate for the insufficient nutritional function and structure of a single oil.

The addition of 3% linseed oil and rapeseed oil to the diet of laying hens increases the content of linolenic acid in the egg yolk, decreases the content of n − 6 PUFAs in the egg yolk (and further decreases are observed with increases in the level of rapeseed oil), and increases the n − 3 PUFA content, improving the structure and composition of fatty acids [11]. Other studies have shown that the addition of PO, coconut oil, corn oil, soybean oil, peanut oil, and linseed oil to feed as an experimentally determined balanced oil significantly increases the egg production rate and significantly improves the dark spot egg rate. Supplementation with this balanced oil does not negatively affect the production performance, routine blood tests, or egg quality of laying hens [195]. In addition, the combined use of vegetable oil and fish oil in the diets of laying hens effectively increases the n − 3 PUFA content in the eggs to produce high-quality eggs [119,189].

### 5.4. Oil Oxidation

In intensive egg production, the oxidation of feed oil is a common factor leading to OS in animals. Due to the different oxidation sources and degrees of oxidation, oxidation products are also complex. According to existing studies, the initial oxidation product of the oil autoxidation reaction is hydrogen peroxide. The UFAs in the oil are oxidized to SFAs but the properties of the primary products are unstable. Under the corresponding oxidation conditions, the degree of oxidation continues to increase. The primary products continue to be oxidized to form secondary products, which mainly include lower alcohols, aldehydes, and ketones with relatively stable chemical properties. In general, MDA, a stable compound, is presumed to be the end product of lipid oxidation. Therefore, the MDA content in oils is used as an indicator of the oxidation level of oils [196,197].

Oxidized oil exerts deleterious effects on animal health and subsequently affects normal physiological functions. First, when an animal ingests oxidized oil through food, oxidation products such as hydroperoxide, MDA, and free fatty acids in the oil will enter the animal’s digestive tract. H_2_O_2_ enters the small intestinal mucosal cells in a freely diffused form and then enters the blood circulation [198,199]. MDA reacts with the compounds in cells to generate reactive oxygen free radicals, and these radicals then induce the peroxidation of UFAs on various biofilms in the cells. Subsequently, the structure of the cells is damaged, and this damage affects various physiological functions and ultimately induces cell apoptosis or death [200]. The free fatty acids in the fat will enter the small intestinal mucosal cells along with other oils in the feed and then enter the blood to form blood lipids and become the oxidized components of blood lipids, which increases the degree of oxidation in the body and the level of OS [201].

The aforementioned hazards of oxidized oils lead to decreased animal health, reduced production performance, such as regarding reproduction, and lower animal product quality. Compared with the results obtained with a control diet, feed containing oxidized oils promotes the synthesis of more mitochondrial enzymes in the animal to eliminate harmful peroxides, which results in increases in the liver size and weight. However, according to indicators such as the antioxidant concentrations in serum, liver, and other tissues, oxidized oils reduce the antioxidant metabolism of the animal body [132]. The content of active oxygen free radicals in the body increases, and these radicals attack intestinal epithelial cells rich in PUFAs, which reduces the barrier and absorption function of the intestinal cell membrane, causes OS in the intestine, and reduces the protein and lipid levels in the gastrointestinal tract. Decreased feed digestibility in the animal’s gastrointestinal tract causes an oxidation reaction in the animal’s ovaries, which reduces the reproductive performance of the animal and the quality of the resulting animal products. In laying hens, oxidized oil significantly reduces the production performance of the hens and destroys the integrity of boiled yolks; the addition of oxidized oil to the diet increases the expression of liver antioxidant-related genes such as glutathione S-transferase (*GST*) and superoxide dismutase 2 (*SOD2*), which indicates that the body is under varying degrees of OS [133]. Therefore, the use of oxidized oils should be avoided in the production of laying hens.

## 6. Conclusions

The effect of oils on the production performance of laying hens is influenced by many factors, such as the type of oil or oils [134], the level added [202], the nutritional level [203], and the breed and physiological status of the laying hens [204]. Consistent with the standards of feed intake, the addition of a low dose of oil does not cause significant fluctuations in the production performance of laying hens, but the addition of a high dose of oil inevitably exerts a greater effect on the production performance and egg quality of the laying hens.

High oil intake increases the metabolic burden of chickens, increases lipid accumulation in the liver, and significantly reduces the production performance of laying hens [205]. Currently, animal and vegetable oils are widely used to improve the fatty acid composition of eggs and the egg quality and have yielded good production benefits. When supplementing the diets of laying hens with animal and vegetable oils, the characteristics of the oils, the feeding conditions, and the supplementation levels should be comprehensively considered to maximize the economic benefits of laying hen production. In addition, balanced oil diets supplemented with oils from different sources have shown excellent potential to improve laying hen performance and egg quality. The mechanism through which dietary oils alter the production performance, egg quality, and body health of laying hens also needs crucial further study.

## Figures and Tables

**Table 1 animals-11-03482-t001:** Studies showing the effects of different oils on egg quality and production performance.

Supplements	Results (Production Performance/Egg Quality/Nutritional Content)	Author/s
Soybean oil	Its addition increases the egg production rate and feed conversion rate; increases the amount of calcium deposited in eggshells and significantly reduces the rate of broken or soft eggs; improves the glycolipid metabolism of laying hens; improves the antioxidant capacity of hens; improves the nutritional value of eggs, reduces the content of cholesterol in eggs, and reduces the TI and AI.	[10,11,76,77,78,79,80]
Rapeseed oil	The addition of 1–5% rapeseed oil has no significant effect on production performance. The addition of 2–6% rapeseed oil reduces egg production, and the addition of 5% rapeseed oil significantly increases the egg yolk weight and contents of DHA and n − 3 FAs in the egg yolk.	[11,12,13,81,82,83,84,85]
Linseed oil	The addition of 2–3% linseed oil does not significantly alter production performance; the addition of 5% or higher concentrations of linseed oil significantly reduces the body weight and egg production rate due to a high n − 3 PUFA deposition efficiency.	[86,87,88,89,90,91,92,93,94]
Palm oil	With increases in the level of red palm oil added (1–3%), the production performance of laying hens improves, the color of the egg yolk significantly improves, and the nutritional value of the egg yolk and the flavor of the egg also improve. The addition of this oil also reduces the TG levels in the egg yolk and increases the MUFA contents in the egg yolk.	[95,96,97,98,99,100,101]
Cottonseed oil	The addition of this oil significantly reduces the egg production rate, average egg weight, and FCR of laying hens, hardens the egg yolk, causes rubbery eggs, and changes the composition of the protein in the yolk granules and plasma.	[102,103,104]
Microalgae oil	The addition of this oil does not negatively affect the production performance and egg quality, increases the ratio of n − 3/n − 6 PUFAs in the egg yolk, and optimizes the FA composition of the egg yolk.	[92,105,106,107,108,109,110,111,112,113,114,115,116,117,118,119,120,121]
Fish oil	Excess fish oil significantly reduces the production performance and egg quality of laying hens and increases the deposition of EPA and DHA in the egg yolk.	[108,122,123,124,125]
Lard	Excess lard reduces the egg production rate and induces fatty liver development but increases the yellow color of the egg and the FA content of the egg yolk.	[32,126,127,128]
Cod liver oil	By promoting the absorption of calcium, it will increase the proportion of unsaturated fatty acids in the egg yolk; poor quality cod liver oil will reduce the quality of eggs, adding too much cod liver oil will make the eggs have a significant fishy smell.	[129,130,131]
Oil oxidation	The addition of oxidized oil reduces the production performance of hens, destroys the integrity of the yolk globules of cooked eggs, and makes animals more susceptible to oxidative stress.	[132,133,134]

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
