# Peer review of "Effect of Oils in Feed on the Production Performance and Egg Quality of Laying Hens"

_animals, 2021, doi:10.3390/ani11123482_

Round 1

Reviewer 1 Report

The paper addresses well-studied issues. It is based on a large number of reports, but some of the issues are described very briefly. I suggest you make corrections before proceeding further with the paper
I suggest that you treat the first paragraph of the abstract as a simple summary. As an abstract, please indicate the most important parameters influenced by oils used in laying hens' diets and include an appropriate summary. Additionally, lines 22-27 are formatted incorrectly. This should be corrected.
Keywords: do not use the words given in the title of the paper! Please change the keywords. For example, the most discussed oils etc. should be used.
Line 34 there is a repetition of the phrase " laying hens" please change the sentence formation to avoid this
Lines 35-36- the sentence in this version makes no sense. If the authors meant to produce energy-rich compound feeds, please use the correct wording.
For eggs, I strongly recommend using the term albumen instead of white. Please change this throughout the manuscript.
Lines 65- 67 indicating differences in specific elements in different feed groups are worth noting e.g. Addition of 6% oil will decrease egg weight (XXX vs YYY)
Line 73- generally assumed that statistically insignificant differences are within P>0.05, brackets are therefore unnecessary
Lines 94-95 the sentence about the role of UFAs in nutritional volue of eggs, does not fit with the rest of the paragraph. Please remove it, or place it elsewhere in the manuscript
Line 108- Internal quality traits are much, much more
Line 112-113- It is not necessary to indicate a formula for determining the number of Haugh units. 
The phrase egg protein is inappropriate. The phrase "albumen" should be used
"white to seep into the yolk" - the phrase is not acceptable! These changes are about the migration of water in accordance between the different elements, based on diffusion. Besides , I do not see the point of raising the issue of raw material freshness in the context of the diet of birds. Please either justify and support your choice with relevant literature, or remove the sections on egg freshness.
Line 141 - please verify the statement that there is no correlation between shell color and egg quality - there are published reports for hens, quail and other species alike.
Line 147 breeding egg? Did you mean hatching eggs? Not all eggs for hatching are breeding eggs!
Chapters 3 and 4 have nothing to do with the title of the paper. The description of quality traits or general factors affecting productivity is only indirectly related to the topic of the paper. They should be reworded according to the subject of the paper, or removed.
Please use the term breeding only when talking about breeding in the sense of genetic progress, in the case of the 255 lines it is more a matter of rearing.
Please indicate clearly if the phrase feed-to-egg ratio refers to FCR? If yes please change it, if not please make it clear.
Animal oils have been described very poorly. I realize that fish oil is one of the basic ones, but there are also others such as cod liver oil. Their importance is relatively minor, but I suggest mentioning them.
In general, the paper is very much lacking in two basic things in my opinion:
Justification for the use of different oils- Enrichment of eggs in polyunsaturated fatty acids. The cheapest commercially available oils can be used to use oils as an energy source, but the current search is to enrich people's diets in a way that maintains a proper n3:n6 ratio.
Furthermore, the authors do not address important issues related to fatty acid profiles and fatty acid indices. It is essential to address even the basic ones like AI, TI or PI-they are crucial not only for the nutritional value of eggs but also affect shelf life.
Furthermore, the data are presented in a way that is of little interest to the reader. Using more tables, would definitely increase the value of the paper. Please rethink this. The one presented at the end of the manuscript is clear and comprehensive. Maybe it would be worthwhile to create a few on e.g. production effects, egg quality and effects on fatty acid profile?

Author Response

Dear Editors and Reviewers:

On behalf of my co-authors, we thank you very much for giving us an opportunity to revise our manuscript, we appreciate editor and reviewers very much for their positive and constructive comments and suggestions on our manuscript entitled “Effect of Oils in Feed on the Production Performance and Egg Quality of Laying Hens”. (ID: 1452651).

Major points that should be addressed are:

Q1.I suggest that you treat the first paragraph of the abstract as a simple summary. As an abstract, please indicate the most important parameters influenced by oils used in laying hens' diets and include an appropriate summary. Additionally, lines 22-27 are formatted incorrectly. This should be corrected.

Additionally, lines 22-27 are formatted incorrectly. This should be corrected.

Answer: Thanks so much for the Review’s comment and suggestion. Following your suggestion, we put the first paragraph of the abstract as a simple summary and the content and format of the abstract was rewritten and reformatted. In addition, we simply added some important parameters of oil to eggs, such as lipid metabolism, nutrition and health, etc. Please see lines 10-32.

Q2.Keywords: do not use the words given in the title of the paper! Please change the keywords. For example, the most discussed oils etc. should be used.

Answer: Thanks for your suggestion. Please see line 33.

Q3.Line 34 there is a repetition of the phrase " laying hens" please change the sentence formation to avoid this.

Answer: Done as requested. Please see line 40.

Q4.Lines 35-36- the sentence in this version makes no sense. If the authors meant to produce energy-rich compound feeds, please use the correct wording.

Answer: Thanks for your suggestion. We deleted this sentence and added a concluding sentence at the end of the paragraph. Please see lines 40-42.        

Q5.For eggs, I strongly recommend using the term albumen instead of white. Please change this throughout the manuscript.

Answer: According to your suggestion, all whites in the full text have been replaced with "albumen".

Q6.Lines 65- 67 indicating differences in specific elements in different feed groups are worth noting e.g. Addition of 6% oil will decrease egg weight (XXX vs YYY)

Answer: Thanks for your suggestion. Please see lines 69-71.

Q7.Line 73- generally assumed that statistically insignificant differences are within P>0.05, brackets are therefore unnecessary

Answer: Done as requested. Please see line 78.

Q8.Lines 94-95 the sentence about the role of UFAs in nutritional volue of eggs, does not fit with the rest of the paragraph. Please remove it, or place it elsewhere in the manuscript

Answer: Thanks for your suggestion. the sentence about "the role of UFAs in nutritional value of eggs" has been placed in chapter 3.3. Please see lines 163-165.

Q9.Line 108- Internal quality traits are much, much more

Line 112-113- It is not necessary to indicate a formula for determining the number of Haugh units.

The phrase egg protein is inappropriate. The phrase "albumen" should be used

"white to seep into the yolk" - the phrase is not acceptable! These changes are about the migration of water in accordance between the different elements, based on diffusion. Besides, I do not see the point of raising the issue of raw material freshness in the context of the diet of birds. Please either justify and support your choice with relevant literature, or remove the sections on egg freshness.

Line 141 - please verify the statement that there is no correlation between shell color and egg quality - there are published reports for hens, quail and other species alike.

Line 147 breeding egg? Did you mean hatching eggs? Not all eggs for hatching are breeding eggs!

Chapters 3 and 4 have nothing to do with the title of the paper. The description of quality traits or general factors affecting productivity is only indirectly related to the topic of the paper. They should be reworded according to the subject of the paper, or removed.

Answer: Thank you for this valuable feedback. Considering that this review is mainly about the mechanism of the effect of oils on egg quality and production performance, adding a review on egg quality and production performance would make the article too lengthy, therefore, we deleted the Chapters 3 and 4. However, the description of the nutritional composition of eggs in Chapter 3 was retained and placed in lines 165-176.

Q10.Please use the term breeding only when talking about breeding in the sense of genetic progress, in the case of the 255 lines it is more a matter of rearing.

Answer: Done as requested. Please see line 225.

Q11.Please indicate clearly if the phrase feed-to-egg ratio refers to FCR? If yes please change it, if not please make it clear.

Answer: Done as requested. all feed-to-egg ratio in the full text have been replaced with FCR.

Q12.Animal oils have been described very poorly. I realize that fish oil is one of the basic ones, but there are also others such as cod liver oil. Their importance is relatively minor, but I suggest mentioning them.

Answer: Thank you for this valuable feedback. According to your suggestion and to supplement the animal oil section, we have added lard and cod liver oil to the usual animal oil section. Please see lines 610-653.

Q13.Justification for the use of different oils- Enrichment of eggs in polyunsaturated fatty acids. The cheapest commercially available oils can be used to use oils as an energy source, but the current search is to enrich people's diets in a way that maintains a proper n3:n6 ratio.

Answer: Thanks so much for the Review’s comment and suggestion. Yes, the addition of oils will increase the unsaturated fatty acid content of egg yolks, and the unsaturated fatty acids omega-3 fatty acids and omega-6 fatty acids will be very beneficial to the human body. Therefore, the article is supplemented with a description of omega-3 and omega-6 in details. Please see lines 177-191.

Q14.Furthermore, the authors do not address important issues related to fatty acid profiles and fatty acid indices. It is essential to address even the basic ones like AI, TI or PI-they are crucial not only for the nutritional value of eggs but also affect shelf life.

Answer: Thanks so much for the Review’s comment and suggestion. For an introduction to fatty acid indices, please see lines 193-195. In addition, we have added a separate section on exponential AI and TI to the review, Please see lines 258-275.

Q15.Furthermore, the data are presented in a way that is of little interest to the reader. Using more tables, would definitely increase the value of the paper. Please rethink this. The one presented at the end of the manuscript is clear and comprehensive. Maybe it would be worthwhile to create a few on e.g. production effects, egg quality and effects on fatty acid profile?

Answer: Thanks so much for the Review’s comment and suggestion. According to your comments, we have reworked the contents of the table, mainly describing the description of feeding oils on production performance, egg quality and nutritional composition. Please see lines 290-291.

Reviewer 2 Report

This is an interesting, and generally well written article about the use of different oils in egg production and in chicken diets. It covers a variety of different oils, and is well referenced throughout.

I only have a few minor comments which are detailed below.

Line 31- delete feed

Line 34, 75 - is this egg feed, or hen feed?

Line 38- have an important ….

Line 41- 42- not completely sure that I follow this. Perhaps consider rewording it

Line 58- delete are oils

Line 59- (TGs) are extracted from ….

 Line 104- please ensure that bacterial species names are in italics

Line 123- the lipid of the egg contains …..

Line 134- nutritional or economic benefits?

Line 226- reference needed

Line 232- is there evidence that this can increase immunity?

Line 246- development of what?

Line 283- please put Latin names into italics

Line 314- delete hen

Line 334- vegetable oil raised in laying hens- please reword this as it doesn’t make sense

Line 336- please add a reference in here

Line 337- better than what?

Line 354-355- please reword as this doesn’t make sense

In the flaxseed section there is a switch from flaxseed to linseed which is difficult to follow

Line 432- this is a repeat of what is above

Palm oil has multiple environmental and conservation issues

Line 473- 475- references needed

Line 486- I think that this would benefit from some further details in here

Line 531- 534- please reword this sentence as it is unclear

Line 537- does not? Rather than dose not?

Line 540 should the plant name be in italics

Line 578- maybe a language difference but is album height correct?

Table 1- I would be tempted to switch this around so that the reference is last rather than first

Author Response

Dear Editors and Reviewers:

On behalf of my co-authors, we thank you very much for giving us an opportunity to revise our manuscript, we appreciate editor and reviewers very much for their positive and constructive comments and suggestions on our manuscript entitled “Effect of Oils in Feed on the Production Performance and Egg Quality of Laying Hens”. (ID: 1452651). 

Major points that should be addressed are:

This is an interesting, and generally well written article about the use of different oils in egg production and in chicken diets. It covers a variety of different oils, and is well referenced throughout. I only have a few minor comments which are detailed below.

Q1. Line 31- delete feed

Answer: Done as requested. Please see line 36.

Q2. Line 34, 75 - is this egg feed, or hen feed?

Answer: According to the proposal, the egg feed that appear in the article have been replaced with hen feed.

Q3. Line 38- have an important ….

Answer: Done as requested. Please see line 43.

Q4. Line 41- 42- not completely sure that I follow this. Perhaps consider rewording it

Answer: Thanks for your suggestion. Please see lines 45-48.

Q5. Line 58- delete are oils

Answer: Done as requested. Please see line 63.

Q6. Line 59- (TGs) are extracted from ….

Answer: Done as requested. Please see line 64.

Q7. Line 104- please ensure that bacterial species names are in italics

Answer: Thanks for your suggestion. Done as requested, Please see lines 107-108.

Q8. Line 123- the lipid of the egg contains …..

Answer: Done as requested. Please see line 165.

Q9. Line 134- nutritional or economic benefits?

Answer: Thanks for your suggestion, we think it would be better to choose the nutritional benefits. Please see lines 174-175.

Q10. Line 226- reference needed

Answer: Done as requested. Please see line 199.

Q11. Line 232- is there evidence that this can increase immunity?

Answer: Thanks for your suggestion we have added the relevant reference about increasing immunity. Please see line 203.

Q12. Line 246- development of what?

Answer: It’s embryonic development. Please see line 216.

Q13. Line 283- please put Latin names into italics

Answer: Done as requested. Please see lines 253-254.

Q14. Line 314- delete hen

Answer: Done as requested. Please see line 305.

Q15. Line 334- vegetable oil raised in laying hens- please reword this as it doesn’t make sense

Answer: Thanks for your suggestion. The sentence was rewritten, Please see lines 324-325.

Q16. Line 336- please add a reference in here

Answer: Done as requested. Please see line 327.

Q17. Line 337- better than what?

Answer: Thanks for your suggestion. The sentence was rewritten, Please see line 328.

Q18. Line 354-355- please reword as this doesn’t make sense

Answer: Done as requested. Please see lines 345-347.

Q19. In the flaxseed section there is a switch from flaxseed to linseed which is difficult to follow

Answer: Thanks for your suggestion. linseed would be more professional in terms of expertise, so replace flaxseed with linseed in the full text.

Q20. Line 432- this is a repeat of what is above. Palm oil has multiple environmental and conservation issues.

Answer: Done as requested, the sentence was deleted.

Q21. Line 473- 475- references needed

Answer: Done as requested. Please see lines 470-471.

Q22. Line 486- I think that this would benefit from some further details in here

Answer: Thank you for this valuable feedback. As recommended, some details on the effect of cotton phenol on egg quality have been added. Please see lines 488-493.

Q23. Line 531- 534- please reword this sentence as it is unclear

Answer: Thanks for your suggestion. The sentence was rewritten, Please see lines 535-537.

Q24. Line 537- does not? Rather than dose not?

Answer: Thanks for your suggestion. dose not has been replaced with does not, Please see line 539.

Q25. Line 540 should the plant name be in italics

Answer: Done as requested. Please see line 541.

Q26. Line 578- maybe a language difference but is album height correct?

Answer: Thanks for your suggestion. All the album height appearing in the article have been replaced with albumen height.

Q27. Table 1- I would be tempted to switch this around so that the reference is last rather than first

Answer: Thanks for your suggestion. As you suggested we have placed the order of references at the end of the table. Please see lines 290-291.

Reviewer 3 Report

Dear authors,

First at all, I want to thank for the possibility to review this manuscript, which contains interesting information for the scientific community since it provides important insights and a very complete bibliographic review about the management of poultry farms. Moreover, this information can be easily transferible to poultry sector.

Nevertheless, I have some issues which I would like to be clarified or rewritten: 

- English grammar and expressions, especially in abstract and introduction, need to be thoroughly reviewed before publication.

- Line 17: “Oils, which are…”.

- Line 24: “egg weight, feed intake, feed egg ratio, and egg quality parameters”.

- Line 25: Add a comma after “yolk color”.

- Line 28: It is not appropriate to use the same words or expressions in the title and in the keywords. Please rewrite it.

- Line 32-33: “feed intake, animal immunity, and…”.

- Line 34: “to egg production feed”.

- Lines 35-36: The word "energy" is repetitive.

- Line 38: Needless to say, eggs are a product of laying hens. It is understood in the context of the manuscript.

- Lines 39-40: The fact that it is an excellent source of protein already implies that it has a high nutritional value.

- Line 42: “Eggs are nutritious” this concept is repetitive.

- Line 45: “Mainly derived from…”.

- Lines 64-65: “reduce the egg production (70.98%), the average egg weight (61.68 g), and the average daily feed intake 65 (109.52 g) …”.

- Line 113: H does not represent protein height. H = albumen height (in millimeters).

- Lines 141-142: This information is incorrect. There are some bibliographic references that indicate that external color coordinate is related to the internal quality characteristic in eggs, for instance:

  • González Ariza, A., Navas González, F. J., Arando Arbulu, A., León Jurado, J. M., Barba Capote, C. J., & Camacho Vallejo, M. E. (2019). Non-parametrical canonical analysis of quality-related characteristics of eggs of different varieties of native hens compared to laying lineage. Animals9(4), 153.

- Line 193-194: The correct denomination is “intermediate-density lipoprotein”, as indicated by its initials IDL.

- Line 278: Delete the point.

- Line 330: “albumen”.

- Lines 344-346: Please add a bibliographic reference.

- Line 635: Please rewrite this sentence thus: “such as albumen weight, yolk weight, albumen height, and therefore, HU”.

- Lines: 379-384: Please add a bibliographic reference.

- Lines 432-433: It would be interesting to include, if any, studies on how the addition of CPO affects the palatability of the feed and if this additive is attractive to hens.

- Lines 477-478: Please could you explain in more detail the process of how the addition of this oil gives rise to rubber eggs?

- Line 483: “albumen”.

Author Response

Dear Editors and Reviewers:

On behalf of my co-authors, we thank you very much for giving us an opportunity to revise our manuscript, we appreciate editor and reviewers very much for their positive and constructive comments and suggestions on our manuscript entitled “Effect of Oils in Feed on the Production Performance and Egg Quality of Laying Hens”. (ID: 1452651).

Major points that should be addressed are:

First at all, I want to thank for the possibility to review this manuscript, which contains interesting information for the scientific community since it provides important insights and a very complete bibliographic review about the management of poultry farms. Moreover, this information can be easily transferible to poultry sector. Nevertheless, I have some issues which I would like to be clarified or rewritten: 

- English grammar and expressions, especially in abstract and introduction, need to be thoroughly reviewed before publication.

Q1. Line 17: “Oils, which are…”.

Answer: Done as requested. Please see line 12.

Q2. Line 24: “egg weight, feed intake, feed egg ratio, and egg quality parameters”.

Answer: Done as requested. Please see line 24.

Q3. Line 25: Add a comma after “yolk color”.

Answer: Done as requested. Please see line 25.

Q4. Line 28: It is not appropriate to use the same words or expressions in the title and in the keywords. Please rewrite it.

Answer: Thanks for your suggestion. Please see line 33.

Q5. Line 32-33: “feed intake, animal immunity, and…”.

Answer: Done as requested. Please see line 37.

Q6. Line 34: “to egg production feed”.

Answer: Thanks for your suggestion. “the to egg feed” in the article has been replaced with to “hen feed”. Please see line 39.

Q7. Lines 35-36: The word "energy" is repetitive.

Answer: Thanks for your suggestion. To make the article content more accurate, the sentence has been rewritten. Please see lines 40-42.

Q8. Line 38: Needless to say, eggs are a product of laying hens. It is understood in the context of the manuscript.

Answer: Thanks for your suggestion. The sentence was rewritten as "Eggs have an important economic value ", Please see line 43.

Q9. Lines 39-40: The fact that it is an excellent source of protein already implies that it has a high nutritional value.

Answer: The sentence was rewritten according to your suggestion. Please see line 44.

Q10. Line 42: “Eggs are nutritious” this concept is repetitive. Please see line

Answer: Done as requested. Please see line 48.

Q11. Line 45: “Mainly derived from…”.

Answer: Done as requested. Please see lines 50-51.

Q12. Lines 64-65: “reduce the egg production (70.98%), the average egg weight (61.68 g), and the average daily feed intake 65 (109.52 g) …”.

Answer: Thanks for your suggestion. Please see lines 69-71.

Q13. Line 113: H does not represent protein height. H = albumen height (in millimeters). Lines 141-142: This information is incorrect. There are some bibliographic references that indicate that external color coordinate is related to the internal quality characteristic in eggs, for instance:

Answer: Thank you for this valuable feedback. Considering that this review is mainly about the mechanism of the effect of oils on egg quality and production performance, adding a review on egg quality and production performance would make the article too lengthy, therefore, we removed the Chapters 3 and 4.

Q14. Line 193-194: The correct denomination is “intermediate-density lipoprotein”, as indicated by its initials IDL.

Answer: Thanks for your suggestion. Please see lines 129-130.

Q15. Line 278: Delete the point.

Answer: Done as requested. Please see line 248.

Q16. Line 330: “albumen”.

Answer: Thanks for your suggestion. All the album height appearing in the article have been replaced with albumen height.

Q17. Lines 344-346: Please add a bibliographic reference.

Answer: Done as requested. Please see lines 336-337.

Q18. Line 635: Please rewrite this sentence thus: “such as albumen weight, yolk weight, albumen height, and therefore, HU”.

Answer: Done as requested. Please see lines 355-356.

Q19. Lines: 379-384: Please add a bibliographic reference.

Answer: Done as requested. Please see lines 373-375.

Q20. Lines 432-433: It would be interesting to include, if any, studies on how the addition of CPO affects the palatability of the feed and if this additive is attractive to hens.

Answer: Thanks for your suggestion. There are not many articles about CPO affecting the palatability of hen diets, but we still found one. Please see lines 427-429.

Q21. Lines 477-478: Please could you explain in more detail the process of how the addition of this oil gives rise to rubber eggs?

Answer: Thank you for this valuable feedback. Specific details about the formation of rubber eggs have been redescribed. Please see lines 472-478.

Q22. Line 483: “albumen”.

Answer: All the album height appearing in the article have been replaced with albumen height.

Round 2

Reviewer 1 Report

The authors have addressed all of the reviewer's comments and the improvements made are, in my opinion, sufficient

Reviewer 3 Report

Dear authors,

All the suggestions I made for the previous version of the paper have been substantially improved. The text has been greatly improved and new bibliographic references have been added to support the new information.

I want to congratulate you for the work done and I think that the manuscript has enough quality and interest to be published in "animals".